# Theranostic Cancer Treatment Using Lentinan-Coated Selenium Nanoparticles and Label-Free CEST MRI

**DOI:** 10.3390/pharmaceutics15010120

**Published:** 2022-12-29

**Authors:** Guanfu Liu, Jiabao Ling, Lizhen He, Yuan Xu, Tianfeng Chen, Changzheng Shi, Liangping Luo

**Affiliations:** 1Department of Radiology, The First Affiliated Hospital of Jinan University, Guangzhou 510632, China; 2Department of Chemistry, Jinan University, Guangzhou 510632, China

**Keywords:** nanotherapeutics, selenium nanoparticles, lentinan, MRI

## Abstract

Selenium nanoparticle (SeNP)-based nanotherapeutics have become an emerging cancer therapy, while effective drug delivery remains a technical hurdle. A theranostic approach, through which imaging companions are integrated with SeNPs, will allow image-guided drug delivery and, therefore, is highly desirable. Traditional methods require the chemical conjugation of imaging agents to the surface of nanoparticles, which may impede the later clinical translation. In this study, we developed a label-free strategy in which lentinan-functionalized SeNPs (LNT-SeNPs) are detected using MRI by the hydroxyl protons carried on LNT molecules. The in vitro phantom study showed that LNT and LNT-SeNPs have a strong CEST signal at 1.0 ppm apart from the water resonance, suggesting an in vivo detectability in the µM concentration range. Demonstrated on CT26 colon tumor cells, LNT-SeNPs exert a strong anticancer effect (IC50 = 4.8 μM), prominently attributed to the ability to generate intracellular reactive oxygen species. However, when testing in a mouse model of CT26 tumors, administration of LNT-SeNPs alone was found unable to deliver sufficient drugs to the tumor, leading to poor treatment responses. To improve the drug delivery, we co-injected LNT-SeNPs and TNF-α, a previously reported drug that could effectively damage the endothelial cells in the tumor vasculature, thereby increasing drug delivery to the tumor. Our results revealed a 75% increase in the intratumoral CEST MRI signal, indicating a markedly increased delivery efficiency of LNT-SeNPs when combined with TNF-α. The combination therapy also resulted in a significantly enhanced treatment outcome, as revealed by the tumor growth study. Taken together, our study demonstrates the first label-free, SeNP-based theranostic system, in which LNT was used for both functional surface coating and CEST MRI signal generating. Such a theranostic LNT-SeNP system is advantageous because it requires chemical labeling and, therefore, has high biocompatibility and low translatable barriers.

## 1. Introduction

Selenium nanoparticles (SeNPs) have drawn increasing interest as a new type of cancer nanotherapeutic thanks to a number of desirable features that SeNPs inherently possess, including highly selective anticancer effect, low systemic toxicity, and versatility for surface modifications and bioconjugation [1]. To further increase the biocompatibility and stability, much effort has been made to coat the surface SeNPs with polysaccharides [2,3,4,5]. For example, Wu et al. reported that SeNPs decorated with mushroom polysaccharide–protein complexes exhibited a significant inhibitory effect on the growth of MCF-7 human breast carcinoma cells [6]. In another recent study, we also showed that SeNPs coated with lentinan, a beta-glucan isolated from the shiitake mushroom (Lentinula edodes), exert potent anticancer effects against hepatocellular carcinoma (HCC) [7]. Interestingly, lentinan (LNT) alone also has anticancer effects, as demonstrated by various preclinical [8,9] and clinical studies [10]. Currently, LNT is used in Japan and China as adjuvant cancer chemotherapy and immunotherapy. In our previous study [7], coating SeNPs with LNT could significantly augment the transport and penetration in HepG2 tumor spheroids and subsequent tumor cell internalization, providing a substantially enhanced inhibition of the proliferation and migration of tumor cells. The synergized effect of LNT and SeNPs was also demonstrated by another recent study in which the anticancer effect of LNT-coated SeNPs was investigated in animal models of Ehrlich ascites cancer (EAC) and OVCAR-3 malignant ascites [11].

Despite the great promise, it should be noted that the effectiveness of LNT-SeNPs is largely determined by the quantities of nanoparticles that can reach the tumor site. Effective drug delivery remains one of the most formidable obstacles for most cancer therapy due to the fact that solid human tumors are biologically heterogeneous, referring to a distinct variation in genotypes and phenotypes, which results in divergent biological behaviors [12,13,14]. For instance, the spatial variation in tumor vascular architecture and the related physiological functioning (perfusion and diffusion) of tissue barriers (i.e., blood–tumor barrier) are responsible for the highly heterogeneous responses of solid tumors to chemotherapies [15,16,17,18,19,20]. When applying nano-chemotherapeutics, where the EPR effect is believed to be the key mechanism via which nanoparticle drug carriers act, tumor heterogeneity is often problematic and leads to unpredictable therapeutic efficacy. Many studies have revealed that the EPR effect in human cancers is rather complex and depends highly on the tumor type, size, and stage [21]. For example, the same ^111^In-labeled PEGylated liposomes (~100 nm) exhibited different biodistribution and pharmacokinetics in patients with different types of locally advanced cancers. [22] Learning from these previous experiences, the development of new LNT-SeNPs therapy would be greatly benefited by a noninvasive, clinically applicable imaging companion that can render and quantify the drug delivery of LNT-SeNPs. The capacity of imaging will be invaluable because it can predict the efficacy of an LNT-SeNP treatment shortly after drug administration and help to develop more robust drug delivery strategies that may substantially augment the tumor uptake of LNT-SeNPs. Among currently available imaging modalities, MRI is the method of choice because of its wide clinical availability, lack of ionic radiation, good soft-tissue contrast, and lack of tissue penetration limitations. In this study, we aim to develop a label-free approach for MRI detection of LNT-SeNPs and use it to monitor the improved drug delivery in a tumor model of low permeability by combining vascular-disrupting therapy. On the virtue of abundant hydroxyl protons carried on LNT molecules, LNT and, thus, LNT-SeNPs can be detected by an advanced MRI technology called chemical exchange saturation transfer (CEST) in a label-free manner (Figure 1). Hence, our study can lead to new theranostic applications of LNT-SeNPs in the precision treatment of cancer.

## 2. Materials and Methods

### 2.1. Chemicals and Reagents

Unless otherwise stated, all chemicals and reagents were purchased from Sigma Aldrich (St. Louis, MO, USA). Lysotracker Red was purchased from Thermo Fisher Scientific (Waltham, MA, USA). Lentinan (LNT) was purchased from Sciphar (Shangluo, China). All in vitro measurements were performed in triplicate.

### 2.2. Synthesis and Characterization of LNT-Coated Selenium Nanoparticles (LNT-SeNPs)

The LNT-SeNPs were synthesized using a modified procedure according to previous publications [7,23]. In brief, 1.6 mL of sodium selenite (Na_2_SeO_3_, 100 mM) and 0.625 mL of LNT (64 mg/mL) were mixed, followed by the dropwise addition of freshly prepared vitamin C (Vc) solution (1.6 mL, 400 mM) under magnetic stirring. The reaction was continued at room temperature in the dark for 12 h and then dialyzed in the dark overnight. The obtained LNT-SeNPs were characterized by a Zeta sizer Nano ZS particle analyzer (Malvern Instruments Co, Worcestershire, UK) and a transmission electron microscope (TEM, JEM-1400Flash; Jeol, Tokyo, Japan). The Se content in LNT-SeNPs was determined by inductively coupled plasma mass spectrometry (ICP-MS, Optima 2000DV, PerkinElmer, Waltham, MA, USA).

### 2.3. Synthesis of Coumarin-6 Labeled LNT-SeNPs and ICG-Labeled LNT-SeNPs

Coumarin-6-labeled LNT-SeNPs and ICG-labeled LNT-SeNPs were synthesized using the same procedure as described above, except that 2 µg/mL coumarin-6 or 1 µg/mL ICG was added slowly in the dark before the Vc solution was added. Unreacted reagents were removed using the same dialysis procedure.

### 2.4. Cells

CT26 mouse colon carcinoma cells (CT26.WT) were purchased from America Type Culture Collection (ATCC, Rockville, MD, USA) and cultured in RPMI-1640 medium supplemented with 10% fetal bovine serum (FBS) using a humidified incubator (37 °C, 5% CO_2_).

### 2.5. Cytotoxicity Assay

CT26 cells were seeded at 2 × 10^3^ cells/well on a 96-well plate and incubated overnight to allow attachment. Cells were then incubated with LNT-SeNPs at concentrations ranging from 1 to 64 µM [Se] for 72 h. Cells cultured under the same conditions but without treatment were used as control. The cell viability was measured by the MTT assay according to the manufacturer’s instruction and the cytotoxicity of LNT-SeNPs was determined by the normalized cell viability at each concentration to that of control. The dose-dependent cell survival curve was fitted using GraphPad Prism 8.0 (GraphPad Software, Inc., San Diego, CA, USA) to estimate the half-maximal inhibitory concentration (IC50).

### 2.6. Measurement of Reactive Oxygen Species (ROS) Generation

The concentration of ROS accumulated intracellularly was measured using the dichloro-dihydro-fluorescein diacetate (DCFH-DA) assay according to previously published procedures [24,25]. In brief, after the incubation of CT26 cells with SeNPs at different concentrations in 96-well plates, 1,3-diphenylisobenzofuran (DPBF, final concentration = 10 μM) was added, followed by measurement using a Versamax fluorescence microplate spectrophotometer (ex/em wavelengths = 488/568 nm) every 10 min for 2 h.

### 2.7. Cell Internalization Analysis

CT26 cells were seeded in 2 cm dishes at a concentration of 30 × 10^4^ cells/mL. After attachment (~12 h), the cells were treated with coumarin-6-labeled LNT-SeNPs for 0, 2, and 24 h. By the end of incubation, cells were washed with PBS to remove the remaining LNT-NPs in cell culture medium. Cells were then collected and measured by flow cytometry (Epics-XL, Beckman, Indianapolis, IN, USA). Additionally, cells were also stained by Hoechst and Lysotracker Red according to the manufacturer’s instructions, and fluorescent images were recorded using a fluorescence microscope (EVOS FL, AMAFD1000, ThermoFisher Scientific, Waltham, MA, USA).

### 2.8. Cell-Cycle Distribution

The effect of the LNT-SeNP treatment on CT26 cell-cycle distribution was analyzed by flow cytometry according to the published procedure [26]. After the incubation with LNT-SeNPs for 72 h, the cells were fixed with 75% ethanol overnight at −20 °C, and then stained with PI at 25 °C for 20–30 min in darkness. The cell-cycle distribution was analyzed using flow cytometry by selecting FSC, SSC, and PE as the channel parameters. Apoptotic cells with hypodiploid DNA content were quantified by the sub-G1 peak in the cell-cycle pattern.

### 2.9. Morphological Changes of Mitochondria

CT26 cells were treated with LNT-SeNPs at the concentrations of 5 μM, 10 μM, or 20 μM, followed by incubation with a mitochondrial membrane potential probe (JC-1) at 37 °C for 30 min [26]. After incubation, the morphological changes were assessed using flow cytometry.

### 2.10. Animals

Animal experiments were performed in accordance with protocols approved by our institutional ethical committee. Tumors allografts were formed by subcutaneously injecting five million CT26 cells into the right flank of Babl/c nude mice respectively (female, 10–11 weeks old, Guangzhou Qingle Life Science Co., Ltd., Guangzhou, China), and then grown for 7–14 days to reach a size (>250 mm^3^) suitable for imaging [27,28]. For in vivo fluorescence imaging, athymic nu/nu (nude) mice (female, 10–11 weeks, Guangzhou Qingle Life Science Co., Ltd., Guangzhou, China) were used.

### 2.11. MRI

All MRI was performed on a 9.4 T Bruker Biospec horizontal scanner (Bruker Biosciences, Billerica, MA, USA). For in vitro CEST MRI characterization, LNT and LNT-SeNPs at different concentrations (1–5 mg/mL) and pH (6.0–8.0) in PBS solutions were measured using a modified RARE-based CEST sequence according to the literature [29] with the following CEST parameters: B1 = 1.2, 2.4, 3.6, 4.7, and 5.9 μT; saturation time (Tsat) = 3 s; offsets ranging from −4 ppm to 4 ppm (step = 0.2 ppm). Imaging acquisition parameters were as follows: TR/effective TE = 6000/43.2 ms, RARE factor = 32, slice thickness = 2 mm, FOV = 15 × 15 mm^2^, matrix size 64 × 64, resolution 0.23 × 0.23 mm^2^, and two averages. The B0 inhomogeneities were measured and corrected using the WASSR method [30].

For in vivo MRI, mice were first anesthetized and positioned inside the MRI scanner. CEST MRI was conducted using the same RARE-based CEST sequence according to previously published protocols [31,32,33], with the following parameters: B1 = 1.8 μT; Tsat = 3 s; offsets ranging from −3 ppm to 3 ppm (step = 0.2 ppm). Imaging acquisition parameters were as follows: TR/effective TE = 5000/3.5 ms, RARE factor = 23, slice thickness = 1 mm, FOV = 30 × 30 mm^2^, matrix size 64 × 64, resolution 0.47 × 0.47 mm^2^, two averages, and total acquisition time = 4.5 min. The B0 inhomogeneities were measured and corrected using the same WASSR method [30]. Mice were first scanned for baseline CEST contrast and then injected with either LNT-SeNPs or LNT-SeNPs + TNF-α intravenously. The doses of LNT-SeNPs (in saline) and TNF-α (10 μg/mL in PBS containing 0.1% (*w*/*v*) BSA) were 500 mg/kg and 100 µg/kg, respectively.

All CEST MRI data processing was performed using custom-written scripts in MATLAB (Mathworks, Waltham, MA, USA). In brief, pixel-wise B0 correction was conducted first. The normalized S^Δω^/S0, where S0 is the water signal intensity without saturation, was plotted as a function of Δω, i.e., the Z-spectrum. MTR_asym_ = (S^−Δω^ − S^+Δω^)/S0 was then computed. To quantify the change in CEST contrast, ΔMTR_asym_ at each timepoint was calculated by MTR_asym_ (t) − MTR_asym_ (pre).

### 2.12. In Vivo and Ex Vivo Fluorescence Imaging

Randomly selected nude mice (n = 3 in each group) were intravenously injected with ICG-labeled LNT-SeNPs (4 mg/kg) alone or together with TNF-α (100 µg/kg). Fluorescence was monitored using an IVIS Lumina Series III (PerkinElmer, Waltham, MA, USA) before and 2 and 24 h post injection. After the last in vivo imaging, mice were sacrificed, and major organs, including heart, lung, liver, spleen, kidneys, and tumor, were harvested and imaged.

### 2.13. Antitumor Effects of LNT-SeNPs

CT26 tumor-bearing mice were randomly divided into four groups to receive treatment of (a) 4 mg/kg LNT-SeNPs, (b) 4 mg/kg LNT-SeNPs + 100 µg/kg TNF-α, (c) 100 µg/kg TNF-α, and (d) 100 µL of saline (vehicle control). Treatments were administrated intravenously every other day for 3 weeks, with injection of TNF-α at intervals of 6 days. Anatomical T2-weighted MRI was performed before and at 3, 7, and 14 days after the first treatment to measure the tumor growth. Tumor volumes were calculated as 0.5 × (length × width^2^) [34,35].

### 2.14. Statistical Analysis

All in vitro experiments were repeated three times. Results were expressed as the mean ± standard deviation (mean ± SD). In vivo data were analyzed using the nonparametric Mann–Whitney U test using Prism 8.0 (GraphPad Inc., San Diego, CA, USA). The level of significance was set at *p* = 0.05.

## 3. Results

### 3.1. Particle Properties of LNT-SeNPs

The successful preparation of spherical LNT-SeNPs was confirmed by TEM images, on the basis of which the average core size of LNT-SeNPs was estimated to be approximately 50 nm (Figure 2A,B). LNT coating significantly increased the hydrodynamic diameter of LNT-SeNPs, i.e., ~131 nm, according to DLS characterization (Figure 2C).

### 3.2. In Vitro Anticancer Effects of LNT-SeNPs

We then systematically evaluated the anticancer effect of LNT-SeNPs on CT26 cancer cells. As shown in Figure 3A, LNT-SeNPs exhibited substantial cytotoxicity against CT26 cells with an estimated IC50 of 4.8 μM. It is well documented that SeNPs exert cytotoxicity in cancer cells by producing a high level of intracellular reactive oxygen species (ROS). To confirm that the observed inhibition effect was correlated with the substantially elevated ROS levels, we measured the intracellular ROS using the DCFH-DA assay. The results showed that LNT-SeNPs induced excessive ROS production in a dose-dependent manner (Figure 3B). At a higher concentration (i.e., >12.5 µM), LNT-SeNPs could result in ~250% increased ROS level as soon as 10 min after the cells were exposed. Even at a low concentration, e.g., 6.25 μM, LNT-SeNPs could induce a ~150% increase in the ROS level at 10 min post exposure and a >200% increase after an extended incubation time (>75 min).

Both the flow cytometric assay and fluorescence microscopic studies revealed that the accumulation of LNT-SeNP internalization was rather quick. The histogram of flow cytometric results in Figure 3C shows that the intracellular fluorescence intensity of coumarin-6 increased with incubation time, indicating continuous internalization of LNT-SeNPs until 8 h when a plateau was reached. The time-elapse microscopic images (Figure 3D), where LNT-SeNPs, lysosomes, and nuclei were stained with coumarin-6 (green), LysoTracker (red), and Hoechst (blue), respectively, show that LNT-SeNPs were quickly taken up by CT26 cells, with a much higher concentration in lysosomes than other cell structures by 1 h, followed by slow lysosomal escape between 1 and 12 h. At 12 h after incubation, the intracellular distribution of LNT-SeNPs was quite uniform. The anticancer effects of LNT-SeNPs on CT26 cells were then confirmed by cell-cycle distribution analysis, in which both G0/G1-phase arrest and apoptotic cell death were observed (Figure 3E). Furthermore, mitochondrial dysfunction was indicated by the marked increase in mitochondrial transmembrane potential by JC-1 staining, e.g., 11.28% vs. 0.51% in the 20 µM LNT-SeNP-treated and control cells, respectively (Figure 3F).

### 3.3. CEST MRI Characteristics of LNT-SeNPs

LNT and, thus, LNT-SeNPs contain abundant hydroxyl protons (Figure 4A), which are rapidly exchanged with surrounding water protons and likely glucose [36] and dextran [31], as previously reported; accordingly, they are able to produce a detectable MRI contrast change using a chemical exchange saturation transfer (CEST) MRI pulse sequence. Figure 4B,C show the CEST contrast of 5 mg/mL LNT (~31.25 mM glucan unit or 10 µM per LNT molecule, MW ~500 kDa according to manufacturer) at different pH values. Similar to glucose [36] or polysaccharides, including dextran [31,32,33] and starch [37], LNT exhibits a broad CEST signal between 0.5 to 2 ppm, with the CEST peak located at approximately 1 ppm at 2.4 µT; the exact peak position shifted with the strengths of the saturation pulse (Figure 4D) due to the fast exchange nature of hydroxyl protons. Figure 4E also reveals that the CEST contrast of LNT is strongly affected by pH, attributed to decreased exchange rates (i.e., from intermediate exchange range to slow exchange range) of hydroxyl protons at lower pH. Using a numerical fitting of experimental data to two-pool Bloch equations [38,39], the exchange rates of LNT were estimated to be 4.2, 3.1, 1.9, 1.6, and 1.5 kHz at a pH of 7.9, 7.4, 7.0, 6.5, and 6.0, respectively. Similarly, LNT-SeNPs also exhibited strong CEST contrast at around 1 ppm (Figure 4F,G). Interestingly, the shape of the Z-spectrum was found to be narrower than that of LNT, indicating that hydroxyl protons had a slower exchange rate on LNT-SeNPs, likely due to the steric hindrance effect among LNT molecules when attached to SeNPs.

### 3.4. CEST MRI Monitoring of LNT-SeNPs Uptake in CT26 Tumors

We then tested the ability of CEST MRI to monitor the tumor uptake of LNT-SeNPs in the CT26 tumors. When LNT-SeNPs were injected alone, as shown in Figure 5, the mean CEST contrast (quantified by MTR_asym_ at 1 ppm) in tumors did not significantly change, i.e., 0.021 ± 0.009 vs. 0.018 ± 0.012 for pre and 1 h post injection, respectively (*p* = 0.50, Mann–Whitney U-test, n = 4). The results indicate negligible uptake of LNT-SeNPs in the tumor, which is consistent with some previous reports [27,40] indicating that the CT26 tumor is poorly permeable to nanoparticles due to the lack of an EPR effect.

An effective and clinically translatable way to increase the drug delivery efficiency in poorly perfused tumors is to augment the EPR using a vascular disrupting agent such as TNF-α [40]. Therefore, we co-injected LNT-SeNPs and TNF-α in another cohort of mice bearing CT26 tumors and conducted CEST MRI. As expected, significantly increased CEST contrast was observed in tumors upon the co-administration, with the mean CEST contrast in tumors increasing by nearly 75%, i.e., 0.020 ± 0.016 vs. 0.035 ± 0.005 for pre and 1 h post injection, respectively (*p* = 0.05, Mann–Whitney U-test, n = 3, Figure 6).

### 3.5. Fluorescence Validation

We then used fluorescence imaging to validate the enhanced tumor uptake of LNT-SeNPs. As shown in Figure 7A, at 2 h post injection, a noticeable fluorescence signal was observed in the tumor receiving LNT-SeNPs + TNF-α but not that receiving LNT-SeNPs only. At 24 h, the fluorescence signal was detected in both groups of tumors, but that in the combination group was much stronger. Moreover, Figure 7B shows the ex vivo fluorescence images, which also confirmed the boosted tumor uptake of LNT-SeNPs by the co-administration of TNF-α. Interestingly, in the co-administration group, stronger uptake in the liver and kidney was also observed, indicating altered pharmacokinetics of LNT-SeNPs due to the effect of TNF-α.

### 3.6. In Vivo Antitumor Activity and Toxicity of LNT-SeNPs

Lastly, we investigated the therapeutic effects of intravenously injected LNT-SeNPs on CT26 tumors using treatment regimens. As shown in Figure 8, only LNT-SeNPs + TNF-α generated significant inhibition of tumor growth after 2 weeks of treatment (tumor size = 332 ± 84 mm^3^ vs. 755 ± 161 mm^3^ in the control group, *p* = 0.0159). No significant tumor growth was observed in the LNT-SeNPs group (tumor size = 610 ± 223 mm^3^, *p* = 0.1905) compared to the control group, nor in the TNF-α group (tumor size = 596 ± 183 mm^3^, *p* = 0.2857).

To better quantify the tumor growth in each individual tumor, we normalized the tumor size in each animal by its starting volume. The results are shown in Figure 8B, where LNT-SeNPs + TNF-α exhibited the strongest inhibition of tumor growth after 2 weeks of treatment (normalized tumor size increase = 7.6 vs. 33.9 in the control group, *p* = 0.0079). The inhibition effect exerted by LNT-SeNPs alone was insignificant (i.e., the normalized tumor size increase = 17.8, *p* = 0.5560). TNF-α alone had no noticeable effect on tumor growth (i.e., the normalized tumor size increase = 35.6, *p* = 0.6905).

## 4. Discussion

In this study, we demonstrated, for the first time, that a functional polymeric coating on the surface of nanoparticles can be utilized as an MRI imaging agent to accomplish theranostic systems without the need for chemical conjugation of metallic or isotopic imaging probes. Despite the great promise of nanoparticle therapeutic systems in treating various preclinical tumor models, as shown by a tremendous number of preclinical studies, to date, only a few nanoparticle therapeutics have been approved by the FDA [41,42,43]. In clinical trials, many nanotherapeutics failed to result in a significant improvement in the overall survival rate of cancer patients. It is now well accepted that human cancer, unlike animal models, has much more heterogenous anatomical and physiological barriers that can hamper effective drug delivery to the targeted tumor cells. In such scenarios, imaging methods that can visualize drug delivery are tremendously useful in predicting the success or failure by detecting whether enough drugs can reach the targeted tumor. As such, image-guided therapeutic systems (also called theranostics), with the integration of imaging and drug delivery systems, are considered to be a promising strategy for accomplishing personalized medicine and significantly improving patient outcomes. However, because chemical labeling of imaging probes is very often needed, the clinical translation of theranostic systems has been slow.

In this study, we chose SeNPs as the model nanoparticles because of their promising anticancer and immunomodulating effects.

One rapidly evolving area for developing SeNPs to be highly effective, clinically applicable anticancer drugs is to develop and optimize suitable surface coatings to improve biocompatibility, stability, and specific tumor targeting. A variety of oligosaccharides, including chitosan [44,45], dextran [46], mushroom polysaccharides [3,6], and protein [4] have been developed for coating SeNPs. Among them, LNT is one of the most attempting coating materials with a clearly demonstrated effectiveness for treating various types of tumor cells or xenografts, such as melanoma [47], hepatocellular carcinoma [7], malignant ascites and ovarian adenocarcinoma [11], lung cancer [48], and colon carcinoma [49], in preclinical studies. However the conventional approach of constructing image-guided or theranostic SeNPs requires chemical conjugation of imaging probes to the surface of nanoparticles, which will increase the complexity of preparation procedures and impose additional components to the system, thus unfortunately hampering future clinical translation substantially.

CEST is an emerging MRI technology in which sensitive and specific detection of diamagnetic compounds is accomplished using MRI in a noninvasive and high-spatial-resolution manner. In the presence of a CEST agent, the MRI signal intensity of water changes, with reference to MRI contrast, when selective saturation pulses are applied to irradiate the exchangeable protons carried by the CEST agents, because these protons can continuously exchange with water to transfer the saturated magnetization [50,51]. There is increasing interest in utilizing CEST MRI to detect biomolecules containing hydroxyl protons, such as glucose [36,52,53] and dextrans [31,33,54,55]. In line with these previous studies, the present study demonstrated the CEST MRI detectability and, thus, theranostic potential of a new polysaccharide LNT. It should be noted that the CEST signal of LNT-SeNPs is pH-dependent, which may be a challenge for accurately quantifying LNT-SeNPs in different microenvironments. However, given that the extracellular pH of tumors is more acidic than that of normal tissue, the sensitivity of CEST MRI detection of LNT-SeNPs is expected to be higher when nanoparticles are located in the extracellular space. Nevertheless, simply using the CEST signal of LNT, we successfully detected the tumor uptake of LNT-SeNPs using noninvasive MRI.

More importantly, we demonstrated the usefulness of theranostics in cancer treatment. As evidenced by our study, CEST MRI may be useful to stratify tumors that are not likely to respond to the treatment because of insufficient drug delivery right after the treatment is given. For example, CT26 tumors are known for the characteristics of the low enhanced permeability and retention (EPR) effect, which exists in many other tumors [56,57,58,59], thus representing a challenging tumor type for nanotherapeutic systems. Qiao et al. showed that co-injection with proinflammatory cytokine tumor necrosis factor-α (TNF-α) could greatly improve the tumor-selective accumulation of liposomes by up to 20-fold [40]. Using this approach, we observed a significantly enhanced CEST contrast in the tumor, indicating substantially improved drug delivery of LNT-SeNPs in this type of tumor.

Our approach, however, had several limitations or technical challenges. First, the frequency offset of hydroxyl protons is close to that of water protons, i.e., around 1 ppm apart. Moreover, a strong CEST background exists at this offset, attributed to a high concentration of endogenous hydroxyl protons. To overcome this challenge, we used a dynamic imaging scheme and continuously monitored the change of CEST MRI signal in the tumor over a 1 h time window post injection. Because animals were kept still inside the scanner, we could directly compare the pre- and post-contrast and calculate the spatially distributed contrast enhancement, similarly to the widely used dynamic contrast enhancement (DCE) MRI. On the basis of this approach, we could accurately assess the contribution of LNT. However, such a long acquisition time may not be feasible in later patient studies. Therefore, new imaging schemes or acquisition methods will have to be designed to accurately determine the post-injection contrast in patients, and a robust co-registration method will be needed to calculate the contrast enhancement in the tumor. Secondly, the doses used in imaging and therapy were actually different in our study. These doses were chosen on the basis of our previous experiences with the effective dosage of LNT-SeNPs on other types of tumors and literature-reported values for CEST MRI [28,60,61]. In the treatment group, LNT-SeNPs were administrated twice-daily for 2 weeks (total injection = 7). Hence, we were concerned that a high-dose regimen may impose too much stress on the mice. In contrast, only a one-time injection was needed in the imaging group. Even though the dose was high, mice showed good tolerance. However, we will investigate the dose effects on both CEST MRI and treatment outcomes in future studies. Lastly, because our approach relies on assessing the contrast changes in the target tissue, such as tumors, correcting motion artefacts will be critical in future applications. In the present study, no noticeable respiration motion artefacts were observed in the tumor region; hence, no motion correction was performed. In our future studies, we will implement a motion correction algorithm to improve the accuracy of MRI measurement in large-cohort animal studies.

## 5. Conclusions

In summary, we demonstrated a label-free approach for detecting LNT-SeNPs by the intrinsic CEST MRI signal of LNT, attributed to the abundant hydroxyl protons existing on the LNT molecules. This new theranostic approach allowed the detection of the uptake of LNT-SeNPs in CT26 colon tumors in mice and monitoring the improved drug delivery in combination therapy. Such a theranostic LNT-SeNP system is much advantageous because no chemical labeling is needed. More importantly, the label-free theranostic strategy can be easily tailored to many other functional biopolymers that have already been used in various nanoparticles to increase biocompatibility and stability or tumor-specific targeting and favorable biodistribution.

## Figures and Tables

**Figure 1 pharmaceutics-15-00120-f001:**
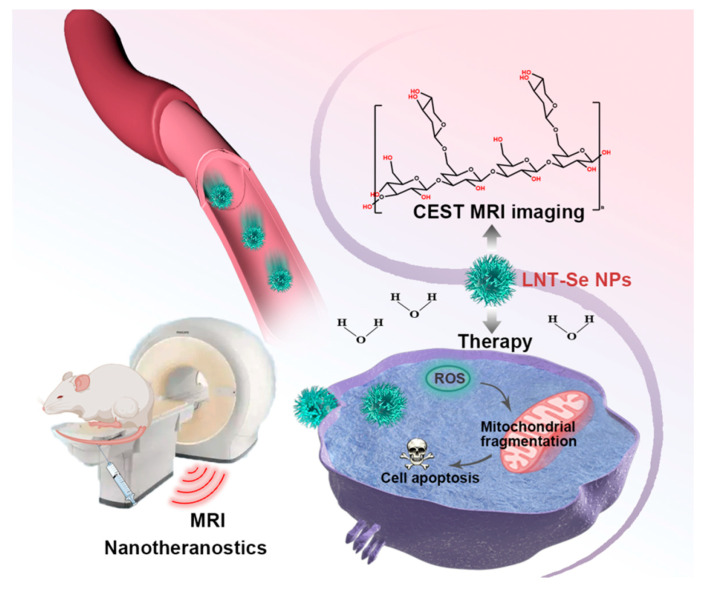
The rational design of theranostic cancer treatment using lentinan-coated selenium nanoparticles and label-free CEST MRI.

**Figure 2 pharmaceutics-15-00120-f002:**
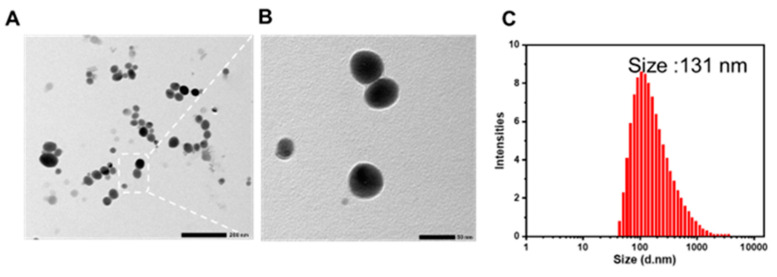
Particle properties of LNT-SeNPs. (**A**,**B**) TEM images of LNT-SeNPs (bar size = 200 and 50 nm, respectively). (**C**) Dynamic light scattering (DLS) measurement.

**Figure 3 pharmaceutics-15-00120-f003:**
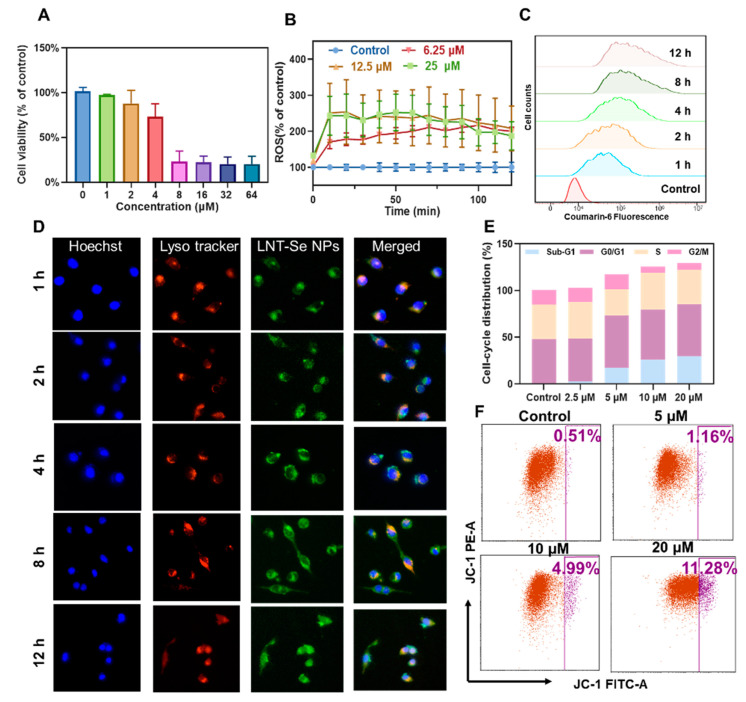
In vitro anticancer effect of LNT-SeNPs. (**A**) Viability of CT26 cells after the incubation of LNT-SeNPs at different concentrations for 72 h, where cells cultured under the same condition but without treatment were used as control. (**B**) Extracellular ROS levels after the incubation of LNT-SeNPs at different concentrations measured by the DCFH-DA assay. (**C**) Flow cytometric assay of CT26 cells after the incubation of LNT-SeNPs after different incubation times. (**D**) Intracellular trafficking of LNT-SeNPs in CT26 cells between 1 to 12 h. (**E**) Cell-cycle distribution of CT26 cells treated with LNT-SeNPs for 72 h. (**F**) Mitochondrial membrane potentials of LNT-SeNPs treated-CT26 cells using the JC-1 flow cytometric assay.

**Figure 4 pharmaceutics-15-00120-f004:**
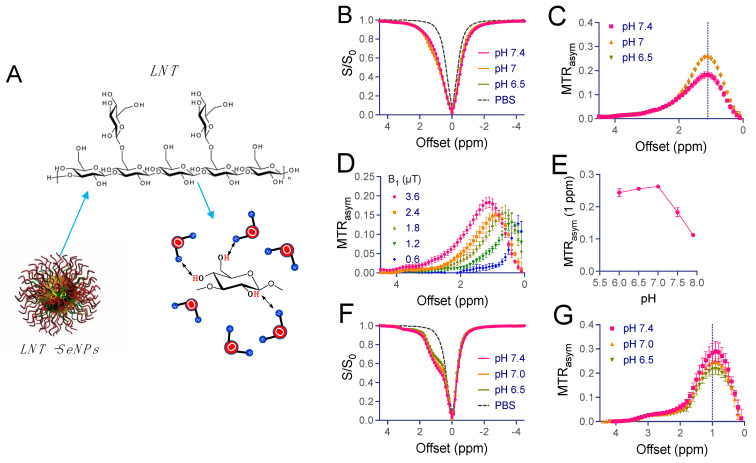
CEST properties of the LNT and LNT-SeNPs. (**A**) Schematic illustration of label-free CEST MRI detection of LNT and LNT-SeNPs by their inherently carried hydroxyl protons. (**B**) Z-spectra and (**C**) MTR_asym_ plots of 3.6 mg/mL LNT (~20 mM glucan) at different pH values. (**D**) Dependency of MTR_asym_ plots on B1 in the range from 0.6 to 3.6 µT. (**E**) pH dependency of the MTR_asym_ values at 1 ppm. (**F**) Z-spectra and (**G**) MTR_asym_ plots of 5 mg/mL LNT-SeNPs at different pH values.

**Figure 5 pharmaceutics-15-00120-f005:**
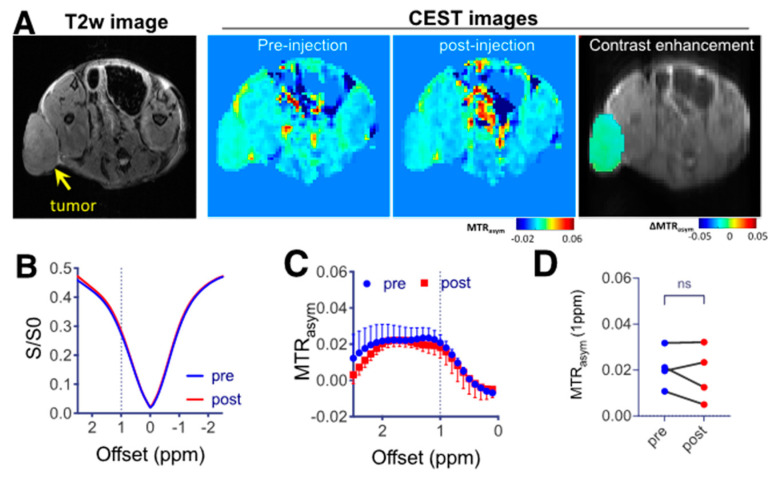
CEST MRI of CT26 tumors in mice receiving LNT-SeNPs. (**A**) T2w image (left), CEST contrast images pre and 1 h post injection of LNT-SeNPs (middle) as quantified by MTR_asym_ at 1 ppm, and an overlay image showing the CEST contrast enhancement in the tumor region on the top of T2w image of a representative mouse. (**B**) Mean Z-spectra and (**C**) mean MTR_asym_ plots of four tumor ROI values. (**D**) Scatter plot of the mean tumor ROI values before and after LNT-SeNP injection (n = 4).

**Figure 6 pharmaceutics-15-00120-f006:**
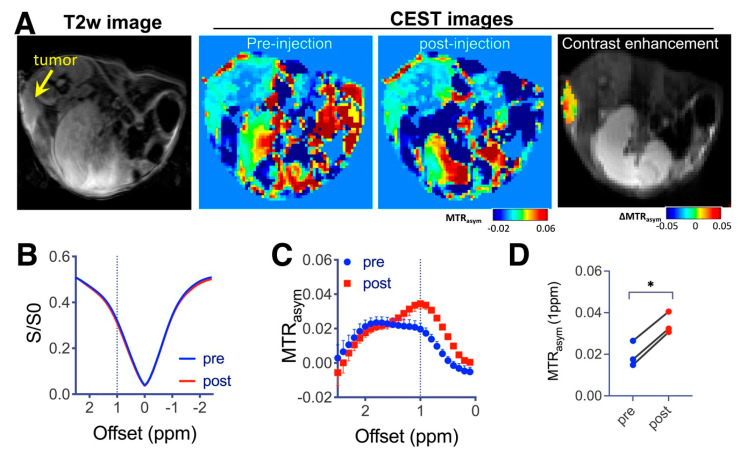
CEST MRI of CT26 tumors in mice receiving co-administration of LNT-SeNPs and TNF-α. (**A**) T2w image (left), CEST contrast images pre and 1 h post injection of LNT-SeNPs and TNF-α (middle) as quantified by MTR_asym_ at 1 ppm, and an overlay image showing the CEST contrast enhancement in the tumor region on the top of T2w image of a representative mouse. (**B**) Mean Z-spectra and (**C**) mean MTR_asym_ plots of three tumor ROI values. (**D**) Scatter plot of the mean tumor ROI values before and after LNT-SeNPs/TNF-α injection (* *p* = 0.05, n = 3).

**Figure 7 pharmaceutics-15-00120-f007:**
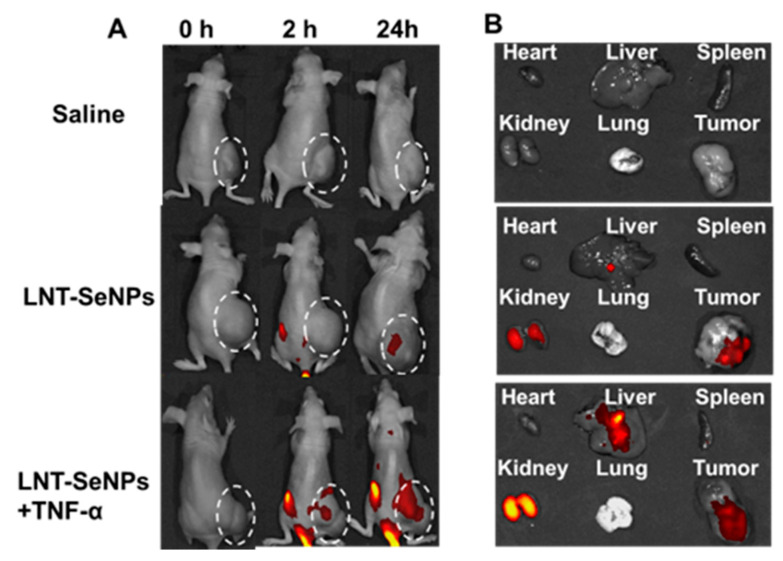
Fluorescence imaging of CT26 tumor-bearing mice receiving different treatments. (**A**) Fluorescence images showing the whole-body distribution of LNT-SeNPs. Tumors were circled by white dashed lines. (**B**) Fluorescence images showing the relative fluorescence intensities among major organs.

**Figure 8 pharmaceutics-15-00120-f008:**
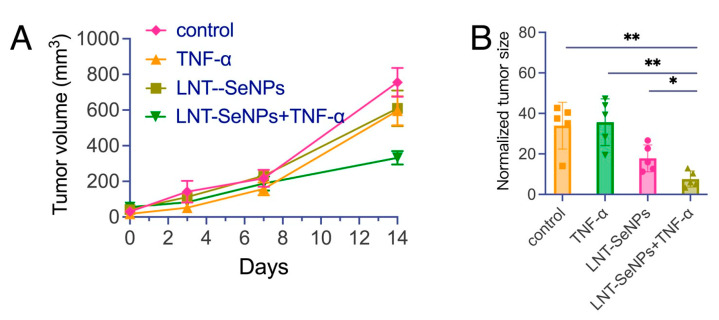
Inhibition of LNT-SeNPs on the growth of CT26 tumors. (**A**) Growth curves of the tumors in mice receiving different treatments. (**B**) Normalized tumor size at 14 days post treatment. All data are presented as the means ± SD (n = 5). Significant differences are indicated as *p* < 0.05 (*) and *p* < 0.01 (**).

## Data Availability

Data presented in this study are available on request from the corresponding author.

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
