# Peer review of "Theranostic Cancer Treatment Using Lentinan-Coated Selenium Nanoparticles and Label-Free CEST MRI"

_pharmaceutics, 2022, doi:10.3390/pharmaceutics15010120_

Round 1

Reviewer 1 Report

 Present manuscript Is well written in which authors have studied the potential of LNT-SeNP in cancer treatment. The following comments need some explanation prior to its acceptance:

Only one cell line CT26 is tested in this study. Will the findings be applicable for all type of solid tumors?

Line 58 "Most cancer therapy" should be "most cancer therapies".

The Material and methods section is unclear. Please include all the information about experimental replicates and controls used.

Figure 3 does not have proper legends. Figure 3 A control what were the experimental conditions for control. was the experiment run in replicates?

Figure 3 A please add the time pint at which the readings were taken.

How was the data analyzed. Did you calculate IC50.

How was the microscopy images signal intensity measured? Any image analysis software could give the more accurate interpretation of this data.

Author Response

R1.1. Only one cell line CT26 is tested in this study. Will the findings be applicable for all type of solid tumors?

Indeed, we expect that the anticancer effects of SeNPs can be generalized to most solid tumor types. For instance, in our previous studies, we have demonstrated that functionalized selenium nanoparticles (SeNPs) exhibit superior antitumor efficacy against human breast cancer cells(Zhang, Biomater Sci, 2019), human cervical carcinoma cells(Luo, Colloids Surf B Biointerfaces, 2012) and lung cancer cells(Wu, J Agric Food Chem, 2013).

In this study, we chose the CT26 tumor model because it is a tumor type known for the characteristics of low enhanced permeability and retention (EPR) effect that exists in many other tumors [56-59], and therefore are a challenging tumor type for nano-therapeutics. In other words, CT26 was chosen not only because it is a type of colorectal tumor but represents hypo-permeable tumors where drug delivery is challenging.

R1.2. Line 58 "Most cancer therapy" should be "most cancer therapies".

Corrected.

R1.3. The Material and methods section is unclear. Please include all the information about experimental replicates and controls used.

We have clarified it in the first paragraph of the Method section: All in vitro measurements were performed in triplicate.

R1.4. Figure 3 does not have proper legends. Figure 3 A control what were the experimental conditions for control. was the experiment run in replicates?

We now replace the “control” with 0 µM to make it clear that cells cultured under the same condition but without treatment were used as control.

Figure 3 A please add the time point at which the readings were taken.

It was measured after a 72-hour incubation.

How was the data analyzed. Did you calculate IC50.

We now added the detail about IC50 estimation using the data shown in Fig. 3A (IC50 of 4.8 μM). We used GraphPad Prism to fit the dose-dependent cell survival curve and determined the IC50 to be approximately 4.8 μM.

How was the microscopy images signal intensity measured? Any image analysis software could give the more accurate interpretation of this data.

We didn’t perform a quantitative analysis of the microscopic images shown in Figure 3D. The purpose of these images was to provide a descriptive analysis of the colocalization of SeNPs inside cells. As for the quantitative analysis, we employed a flow cytometric assay, which is supposed to be more accurate.

Reviewer 2 Report

The article by Liu et al. discusses the theranostic effect of LNT-SeNPs in cellulo and in vivo. The biological studies are made on CT26 cell-based tumors. Due to the poor penetration of LNT-SeNPs into the tumor microenvironment, the authors use TNF-alpha to enhance their penetration and thus take advantage of the theranostic properties of LNT-SeNPs.
In general, can the authors explain the importance of Se nanoparticles  compares to nanoparticles labelled for radiotherapy (hafnium oxide, gadolium) or hyperthermia (gold or iron oxide nanoparticles). These two techniques have very good anti-cancer effects with, in some cases, tumor resorption, so we do not clearly understand the contribution of selenium nanoparticles in this type of application.
More specifically, can the authors answer the following questions:
Line 49-50 "coating with LNT" what is coated to LNP?
Line 223-224 "followed by immersion in the cytoplasm over time". What is the evidence to suggest that the nanoparticles are in the cytoplasm?
Figure 3c does not provide a clear appreciation of the internalization kinetics of nanoparticles.
In the discussion section LNT-SeNPs are described as drug delivery systems, yet there is no drug that is encapsulated or delivered.

Author Response

R2.1. In general, can the authors explain the importance of Se nanoparticles compares to nanoparticles labelled for radiotherapy (hafnium oxide, gadolium) or hyperthermia (gold or iron oxide nanoparticles). These two techniques have very good anticancer effects with, in some cases, tumor resorption, so we do not clearly understand the contribution of selenium nanoparticles in this type of application.

We thank the reviewer for raising this good question and totally agree that there may be a wide array of biomedical applications for SeNPs. We have added the following paragraph to the Discussion. 

Selenium is an essential trace element for the body with low toxicity to normal cells. SeNPs are considered a potential anticancer agent by their ability to inhibit the proliferation rate of tumor cells. Therefore the mechanism of SeNPs is essentially different from other nanoparticles, for instance, hafnium oxide NP and gadolinium NP, which are being investigated in clinical trials as radio-enhancer nanoparticles (Scher, Biotechnol Rep (Amst), 2020). Importantly, the anticancer effects of SeNPs are complementary to radiotherapy and hypothermal therapy. As evidenced by a few recent studies, much stronger anticancer effects could be achieved by combining SeNPs with other therapies. For example, Du et al. developed a poly(vinylpyrollidone)- and selenocysteine-modified Bi2Se3 nanoparticles (PVP-Bi2Se3@Sec NPs) for simultaneously enhancing radiotherapeutic effects and reducing the side-effects of radiation(Du, Adv Mater, 2017). In another study, a triple combination therapy (chemotherapeutic, chemoprevention, and photoablation/hyperthermia) was developed by using Se@Au@mSiO2/DOX nanoparticles, which exhibited an enhanced anticancer efficacy for treating metastatic breast cancer(Ramasamy, NPG Asia Materials, 2018). Our recent study also demonstrated the synergic use of SeNPs and doxorubicin (DOX) and indocyanine green (ICG) as combined chemo-photothermal Therapy (Fang, Chem Asian J, 2018). Moreover, studies also show SeNPs are potential enhancers for cancer immunotherapy (Liu, Nano Today, 2020; Song, Adv Healthcare Mater, 2021). These studies collectively imply the great potential of SeNPs in other applications, including immunotherapy, radiotherapy, and hyperthermal therapy

Refs:

Scher N, Bonvalot S, Le Tourneau C, Chajon E, Verry C, Thariat J, Calugaru V (2020). "Review of clinical applications of radiation-enhancing nanoparticles." Biotechnol Rep (Amst) 28: e00548.

Liu T, Xu L, He L, Zhao J, Zhang Z, Chen Q, Chen T (2020). "Selenium nanoparticles regulates selenoprotein to boost cytokine-induced killer cells-based cancer immunotherapy." Nano Today 35.

Song Z, Luo W, Zheng H, Zeng Y, Wang J, Chen T (2021). "Translational Nanotherapeutics Reprograms Immune Microenvironment in Malignant Pleural Effusion of Lung Adenocarcinoma." Adv Healthcare Mater 10(12): 2100149.

Du J, Gu Z, Yan L, Yong Y, Yi X, Zhang X, Liu J, Wu R, Ge C, Chen C, Zhao Y (2017). "Poly(Vinylpyrollidone)- and Selenocysteine-Modified Bi(2) Se(3) Nanoparticles Enhance Radiotherapy Efficacy in Tumors and Promote Radioprotection in Normal Tissues." Adv Mater 29(34).

Ramasamy T, Ruttala HB, Sundaramoorthy P, Poudel BK, Youn YS, Ku SK, Choi H-G, Yong CS, Kim JO (2018). "Multimodal selenium nanoshell-capped Au@mSiO2 nanoplatform for NIR-responsive chemo-photothermal therapy against metastatic breast cancer." NPG Asia Materials 10(4): 197-216.

Fang X, Li C, Zheng L, Yang F, Chen T (2018). "Dual-Targeted Selenium Nanoparticles for Synergistic Photothermal Therapy and Chemotherapy of Tumors." Chem Asian J 13(8): 996-1004.

R2.2. More specifically, can the authors answer the following questions:
Line 49-50 "coating with LNT" what is coated to LNP?

To make it clear, we rephrased it to

Coating SeNPs with LNT could significantly augment their transport and penetration in HepG2 tumor spheroids and subsequent tumor cell internalization,

R.2.3. Line 223-224 "followed by immersion in the cytoplasm over time". What is the evidence to suggest that the nanoparticles are in the cytoplasm?
Figure 3c does not provide a clear appreciation of the internalization kinetics of nanoparticles.

We agree with the reviewer that SeNPs accumulated in the cytoplasm (outside lysosomes) were not directly evidenced by Figure 3c. What can be directly extracted from the time-elapse microscopic images (where lysosomes and nuclei were stained with LysoTracker and Hoechst, respectively) are 1) coumarin-6-labeled LNT-SeNPs were quickly taken up by CT26 cells, with a much higher concentration in lysosomes than other cell structures by 1 hour, 2) LNT-SeNPs gradually escaped from lysosomes but remained in cells between 1 and 12 hours, 3) By 12 hours, the intracellular distribution of LNT-SeNPs were quite uniform.  We now added these interpretations to the corresponding section as:

The time-elapse microscopic images (Fig. 3D), where LNT-SeNPs, lysosomes, and nuclei were stained with coumarin-6 (green), LysoTracker (red), and Hoechst (blue), respectively, show that LNT-SeNPs were quickly taken up by CT26 cells, with a much higher concentration in lysosomes than other cell structures by 1 hour, followed by slow lysosomal escape between 1 and 12 hours. At 12 hours after incubation, the intracellular distribution of LNT-SeNPs was quite uniform.

We apologize that the axis in Figure 3C was mislabeled, which should be coumarin 6 instead of FITC. Figure 3C shows the fluorescence signal of coumarin 6 increases with incubation time, indicating continuous internalization of LNT-SeNPs till 8 hours when a plateau was reached. We agree with the reviewer that the result reflects the internalization kinetics but is not a direct measurement of the kinetics. We now made it clear in the revised manuscript.

R.2.4. In the discussion section LNT-SeNPs are described as drug delivery systems, yet there is no drug that is encapsulated or delivered.

We now rephrased the drug delivery system (DDS) to therapeutic system or theranostic system in the corresponding sections.

Reviewer 3 Report

The articles describes the anticancer application of complex system of lentinan-coated selenium nanoparticles and label-free CEST MRI. The topic of the article is new and promising. The article is well written and easy to read and understand. The results are well discussed and the conclusions are supported by the data. As for English, I am not native speaker, for me English is acceptable. Hovewer, I have some remarks that are important for this study.

Remarks:

The major drawback of this study is the absence of correct statistical analysis of the data.

The data in vivo in mice were analised by parametric analysis . For n=3 or n=5 the non-parametric analysis of the data should be used.

The most funny that at Fig. 8 B the data are presented as non-parametric - median and 25%/75% procentiles - while the description of Fig. indicate that the data are parametric - mean+-SD.

After the correct statistical analysis its description should be included in Materials and Methods. 

Author Response

R3.1. The major drawback of this study is the absence of correct statistical analysis of the data.The data in vivo in mice were analised by parametric analysis . For n=3 or n=5 the non-parametric analysis of the data should be used.

We thank the reviewer for the suggestion and have replaced the statistical analyses using the Mann Whitney U test for CEST MRI and treatment evaluation.

R.3.2. The most funny that at Fig. 8 B the data are presented as non-parametric - median and 25%/75% procentiles - while the description of Fig. indicate that the data are parametric - mean+-SD.

We apologize for the omission and have updated Figure 8B with the appropriate plot type in which data are shown as mean+/- SD.

 R3.3. After the correct statistical analysis its description should be included in Materials and Methods. 

We have added a section of “Statistical analysis”.

Round 2

Reviewer 3 Report

Everything is O.K. in the revised version.